# Transcranial Direct-Current Stimulation Does Not Affect Implicit Sensorimotor Adaptation: A Randomized Sham-Controlled Trial

**DOI:** 10.3390/brainsci12101325

**Published:** 2022-09-29

**Authors:** Huijun Wang, Kunlin Wei

**Affiliations:** 1School of Psychological and Cognitive Sciences and Beijing Key Laboratory of Behavior and Mental Health, Peking University, Beijing 100871, China; 2Key Laboratory of Machine Perception (Ministry of Education), Peking University, Beijing 100871, China

**Keywords:** motor adaptation, motor learning, cerebellum, transcranial direct current stimulation, non-invasive brain stimulation

## Abstract

Humans constantly calibrate their sensorimotor system to accommodate environmental changes, and this perception-action integration is extensively studied using sensorimotor adaptation paradigms. The cerebellum is one of the key brain regions for sensorimotor adaptation, but previous attempts to modulate sensorimotor adaptation with cerebellar transcranial direct current stimulation (ctDCS) produced inconsistent findings. Since both conscious/explicit learning and procedural/implicit learning are involved in adaptation, researchers have proposed that ctDCS only affects sensorimotor adaptation when implicit learning dominates the overall adaptation. However, previous research had both types of learning co-exist in their experiments without controlling their potential interaction under the influence of ctDCS. Here, we used error clamp perturbation and gradual perturbation, two effective techniques to elicit implicit learning only, to test the ctDCS effect on sensorimotor adaptation. We administrated ctDCS to independent groups of participants while they implicitly adapted to visual errors. In Experiment 1, we found that cerebellar anodal tDCS had no effect on implicit adaptation induced by error clamp. In Experiment 2, we applied both anodal and cathodal stimulation and used a smaller error clamp to prevent a potential ceiling effect, and replicated the null effect. In Experiment 3, we used gradually imposed visual errors to elicit implicit adaptation but still found no effect of anodal tDCS. With a total of 174 participants, we conclude that the previous inconsistent tDCS effect on sensorimotor adaptation cannot be explained by the relative contribution of implicit learning. Given that the cerebellum is simultaneously involved in explicit and implicit learning, our results suggest that the complex interplay between the two learning processes and large individual differences associated with this interplay might contribute to the inconsistent findings from previous studies on ctDCS and sensorimotor adaptation.

## 1. Introduction

When a proficient movement was perturbed by environmental changes, such as playing ping pong with a borrowed, heavier paddle, motor errors would occur. The sensorimotor system seeks to counter the influence of external perturbations, reduce or even eliminate error, and return to its former performance level. This error-reduction process is referred to as sensorimotor adaptation, which has been long regarded as cerebellar-dependent. As supporting evidence, brain imaging studies have found a correlation between cerebellum activity and adaptation [1,2,3]. Patients with cerebellum degeneration show impaired adaptation in different motor tasks [4,5,6,7]. Researchers have also been striving to establish the causal relationship between cerebellar excitability and sensorimotor adaptation using non-invasive brain stimulation such as transcranial direct current stimulation (tDCS). However, the findings are far from conclusive.

Cerebellar tDCS (ctDCS) is capable of modulating cerebellar excitability [8], but its effect on sensorimotor adaptation is inconsistently reported. Galea et al. applied anodal ctDCS to increase cerebellar excitability when people were adapting to rotated visual feedback of the hand (visuomotor rotation). They found that the anodal ctDCS accelerated adaptation and the subsequent re-adaptation to the same perturbation [9]. The faster initial adaptation is viewed as an online effect of ctDCS, while the latter faster re-adaptation, namely savings, reflects the long-term retention of adaptation. This study has inspired continuing efforts to study the relationship between ctDCS and adaptation. Some later studies replicated the online effect of ctDCS on adaptation to different perturbations, including visuomotor rotation [10,11], force field [12], split-belt walking [13], and saccadic adaptation [14]. However, quite a few studies reported a null effect. For example, Taubert et al. found that anodal ctDCS impaired both initial adaptation to a novel force field and the re-adaptation the next day [15]. Mamlins et al. found no effect of anodal or cathodal ctDCS on adaptation to visuomotor rotation or the force field [16]. Specifically, for visuomotor rotation, Galea et al. did not replicate their own seminar study: the effect of ctDCS was tested using a series of experiments, which differed in task specifics, including the orientation of the visual display, timing of tDCS, and perturbation schedule [17]. The previously reported accelerated adaptation was not observed in six of the seven experiments. The only experiment reporting a positive anodal ctDCS effect was not replicated in one of the six failed experiments with an identical task and experimental design. Other researchers also returned inconsistent findings on visuomotor rotation and indicated that the presence of the ctDCS effect depends on the effector used [18]) and the age of participants [19]).

Recent theorization of sensorimotor adaptation offers a new perspective on studying the role of cerebellar stimulation in adaptation. Previous ctDCS studies viewed sensorimotor adaptation as a single procedural process, wherein the cerebellum recalibrates the sensorimotor control in an implicit way. However, recent behavioral and modeling studies have shown that sensorimotor adaptation, and visuomotor rotation adaptation in particular, involves multiple learning components beyond cerebellum-based implicit learning, including reinforcement learning, use-dependent plasticity, and explicit strategy [20,21]. Explicit strategy, or explicit learning, refers to the cognition-based motor strategy to counter perturbations that are consistently present [22]. Its cognitive and explicit nature is evidenced by the findings that the contribution of explicit learning increases with better awareness of the perturbation [23], higher working memory capacity [24], more movement preparation time [25,26], and younger age [27]. Hence, sensorimotor adaptation has a strong cognitive component, which is not readily accounted for by cerebellum-based procedural learning alone [28].

Researchers thus sought to test whether the inconsistent cerebellar tDCS effect was caused by the failure or neglect of separating explicit learning and implicit learning during sensorimotor adaptation. Liew et al. quantified explicit and implicit learning during visuomotor rotation adaptation and found a marginally significant anodal ctDCS effect on implicit learning only when the visual display was vertical [29]. The authors interpreted this finding as supportive evidence that the cerebellar stimulation affected adaptation only when implicit learning was more involved since the vertical display invoked greater implicit learning than the horizontal display, which did not induce a ctDCS effect. Leow et al. found that anodal ctDCS increased adaptation and its aftereffect when the preparation time for movement was hastened [30]. Since the shortening of preparation time would suppress explicit learning, the authors concluded that ctDCS would improve sensorimotor adaptation when implicit learning plays a prominent role in adaptation. However, Leow et al.’s findings also have alternative explanations. First, given the fact that the task was demanding with stringent requirements on the preparation time, ctDCS might facilitate motor preparation and indirectly lead to improved adaptation instead of enhancing implicit learning per se. Furthermore, recent studies have shown that explicit learning is unlikely to be fully suppressed in the conventional visuomotor rotation paradigm: since the perturbation was abruptly applied with salient motor errors, people can still develop cognitive or conceptual learning despite this part of learning being suppressed by a shortened preparation time [31].

Here, we sought to directly test the effect of ctDCS on implicit adaptation by using the error clamp paradigm, a variant of traditional visuomotor rotation that has been shown to be driven solely by implicit learning [32]. Instead of showing rotated feedback of one’s hand, the error clamp paradigm shows a cursor moving in synchrony with the hand along a straight line angularly deviated from the desired movement trajectory. Participants are fully aware that the rotated feedback is “clamped” and should be ignored, but their hands will gradually drift away from the desired trajectory, exhibiting implicit adaptation to the clamped feedback. In the first experiment, we applied cerebellar anodal or sham tDCS during the error clamp learning and examined implicit adaptation and its retention within a day and across days. In the second experiment, we applied both anodal and cathodal ctDCS over the cerebellum during the error clamp learning and also reduced the clamp size to avoid a possible ceiling effect that could have confounded the results in the first experiment. In the third experiment, we used gradually induced visuomotor rotation, another sensorimotor paradigm that is dominated by implicit learning. For a preview, across the board experimental conditions (*n* = 174) that isolated implicit learning, cerebellar tDCS failed to convincingly modulate adaptation and its retention, highlighting the need to re-examine the effect of non-invasive stimulation on the cerebellum and theorization of cerebellar involvement in sensorimotor adaptation.

## 2. Methods

### 2.1. Participants

One hundred and seventy-four right-handed (self-reported) healthy individuals without a history of neurological or psychiatric conditions (97 females; age 21.19 ± 2.56 years, range 18–30 years) participated in the study. Based on the sample size and effect size in two similar studies on cerebellar tDCS and visuomotor rotation, we aimed for a minimum sample size of 10 participants for each group to detect a significant group difference with a power of 0.8 at the significance level of 0.05 [9,29]. In Experiment 1, both the anodal tDCS and sham tDCS groups had 41 participants (20 females, age 20.40 ± 2.23 for the anodal tDCS group; 28 females, age 20.71 ± 2.32 for the sham group). In Experiment 2, the anodal tDCS group had 14 participants (6 females, age 19.23 ± 2.28), the cathodal tDCS group had 14 participants (10 females, age 19.71 ± 2.02), and the sham tDCS group had 10 participants (8 females, age 20.50 ± 4.52). In Experiment 3, the anodal tDCS group had 27 participants (13 females, age 21.11 ± 2.24), and the sham tDCS group had 27 participants (12 females, age 21.81 ± 2.63). All the participants were naïve to the purpose of the experiment, signed an informed consent approved by the Institutional Review Board of Peking University, and received monetary compensation for their participation.

### 2.2. Experimental Setup

The participant was seated in front of a custom-made setup that included a 19-inch LCD screen and a digitizing tablet (Wacom ptk-1240; Figure 1A). The screen was vertically placed with its center at eye level. The tablet was horizontally placed on the desk in front of the participant. A plastic board was placed horizontally above the digitizing tablet at about the shoulder level to prevent the vision of the arm. The room for the experiment was dimly lit throughout.

tDCS was set at 2 mA and controlled by a tDCS low-intensity stimulator (Soterix Medical, Model 1300A). It was delivered through 2 sponge electrodes (surface area: 35 cm^2^) soaked in saline. Two electrodes were tied to the participant’s head with two rubber bands, with one electrode placed 3 cm right of and 1 cm below the inion, and the other positioned on the right buccinator muscle. Depending on the experimental group, the polarity of the electrodes was either anodal or cathodal for the stimulation groups; the anodal and cathodal electrodes were counterbalanced for the sham groups. For the anodal and cathodal groups, the stimulation lasted 20 min. At the onset of tDCS, the current ramped up from 0 to 2 mA over a period of 15 s, was kept at 2 mA for 20 min, then ramped down at the end of the stimulation. For the sham groups, the current ramped up and down at the same rate as in the stimulation groups but without the 20-min stimulation period. Therefore, the sham groups received tDCS for a total duration of 60 s. This procedure was in keeping with other cerebellar tDCS studies [33,34] and has been found to effectively modulate cerebellar excitability as evidenced by changes in cerebellar brain inhibition [8,35,36]. By using similar ramping up and down, all participants were blinded as to whether anodal, cathodal or sham tDCS was being applied, and this was re-confirmed by post-experiment briefing.

### 2.3. Sensorimotor Adaptation Task

The sensorimotor adaptation task here used classical visuomotor rotation perturbation, which required people to adapt to a rotated visual representation of the hand [37]. The participants performed center-out shooting movements with a digitizing pen held by their dominant hand. The pen tip, motion-tracked by the tablet, could be spatially mapped and visualized on the vertical screen as a cursor in real-time. The participants were required to control the cursor to perform a straight-line shooting movement to “slice” through a visual target. Each shooting movement was made from a starting position towards a target. The starting position was indicated by a green dot (10 mm diameter) in the middle of the screen, and the target was indicated by a white dot (6 mm diameter) positioned on a 5 cm radii invisible circle centered around the starting position. The target location at the laterally right of the body midline was defined as 0° and laterally left as 180°. The target could appear at eight possible locations, i.e., 0°, 45°, 90°, 135°, 180°, 225°, 270°, and 315°.

To initiate a trial, the participant had to stay at the starting position to wait for the trial to start. Once the pen tip stayed within 2 cm of the starting position, a solid green cursor (4 mm diameter, the hand cursor) would appear, reflecting the actual position of the hand. After staying for 0.5 s, the target appeared, the hand cursor was turned off, and a sound was simultaneously played as a movement trigger. The participant was required to slice through the target as quickly and as accurately as possible. After the radial distance of movement exceeded 5 cm, the cursor remained where it passed through the invisible circle of 5 cm for an additional 0.5 s, then disappeared with the target. To prevent online correction and promote feedforward learning, we also required the participant to make a fast movement: if the time between the target appearance and its attainment (the movement surpassed 5 cm) were longer than 0.8 s or shorter than 0.2 s, a warning text of “too slow” or “too fast”, respectively, would be displayed along with an unpleasant buzzing sound.

In each of four types of trials, how the cursor feedback was delivered during the shooting movement differed (Figure 1B). In the veridical trials, the cursor represented the actual position of the hand. In the no-feedback trials, there was no visible cursor during movement. Visuomotor rotation perturbation was implemented in the remaining two types of trials. For the error clamp trials, the cursor was restricted to an invariant straight trajectory with an angular offset towards the target. The angular offset stayed invariant against the actual hand movement directions; thus, task-irrelevant feedback “clamped” the error size. The participants were fully aware of this offset and required to ignore the cursor and move straight to the target. In the visuomotor rotation trials, the cursor was also moving with an angular offset, which was rotated against the actual hand movement direction. Thus, this rotated feedback was task-relevant, and the participants could actively rotate their movement direction to move the cursor towards the target.

### 2.4. Experimental Design

Experiments 1, 2, and 3 were all randomized, single-blinded, sham-controlled trials. For each experiment, participants were randomly assigned to different groups with a computer-generated randomization sequence. The study protocol was not pre-registered.

#### 2.4.1. Experiment 1: The Effect of ctDCS on Error Clamp Adaptation and Its Cross-Day Retention

In Experiment 1, we aimed to examine whether ongoing ctDCS could affect implicit sensorimotor adaptation and its long-term retention. Experiment 1 consisted of two sessions, separated into two days to enable the testing of cross-day retention of implicit adaptation (Figure 1C). On each day, there were 3 phases of the experiment, including the baseline, adaptation, and decay phases. Trials in each phase were organized in cycles, each of which included eight movements to each of the 8 possible targets (0°, 45°, 90°, 135°, 180°, 225°, 270°, and 315°). The order of the target appearance was randomized within a cycle. The baseline, learning, and decay phases had 15, 40, and 40 cycles, respectively, resulting in a total of 95 cycles and 760 trials. We applied ctDCS during the adaptation phase on day 1 only and examined whether it affected the implicit adaptation on day 1 and its retention affected the subsequent re-adaptation of classical visuomotor rotation on day 2. In the first 5 cycles of the baseline phase on day 1, the cursor was visible during movement, reflecting the actual hand position. In the next 10 baseline cycles, the cursor would become invisible once moving out of the starting position. The performance of these no-feedback trials was used as a performance baseline for the shooting movement. In the subsequent adaptation phase, we applied the error clamp perturbation, i.e., the cursor was rotated 30° counterclockwise (CCW) against the “desired” trajectory, which was the straight line connecting the starting position to the target. Though the rotated cursor was synchronized with the hand motion, its movement direction did not depend on the hand motion. Although the participants were instructed to ignore this “clamped” feedback and move their hand directly to the target, they would gradually adapt to this cursor rotation by moving the hand clockwise without even knowing it. This adaptation to task-irrelevant feedback without conscious control was regarded as implicit [32]. In the subsequent final decay phase, we withdrew the cursor feedback and examined how the implicit adaptation decayed passively without new visual inputs. Critically, ctDCS was applied on day 1 only to cover the duration of initial adaptation. It was turned on starting from the second half of the no-feedback baseline (cycle 11) and lasted 20 min, a duration covering the whole adaptation phase. Experiment 1 lasted approximately 1.5 h on day 1, including the familiarization with the experimental setup and preparation of tDCS.

On day 2, the participants experienced the same three phases of trials to examine their cross-day retention, lasting about 1 h. All the procedures remained the same as on Day 1, except that the re-adaptation phase used visuomotor rotation trials that involved task-relevant feedback: the cursor was similarly rotated 30° CCW, but it was relative to the actual hand movement. Hence, compared to the error clamp adaptation on day 1, the visuomotor rotation of the same 30° here was task-relevant, and the participant was required to actively compensate for this rotation and bring the cursor, instead of the hand, to the target. The purpose of testing this visuomotor rotation adaptation was to examine whether the potential ctDCS effect on implicit adaptation could manifest itself in retention and improve the conventional sensorimotor adaptation ability across days.

#### 2.4.2. Experiment 2: Smaller Error Clamp and tDCS Polarity

In Experiment 2, we aimed to upregulate and downregulate the excitability of the cerebellum by using anodal and cathodal ctDCS, respectively. Though the cathodal tDCS showed an inconsistent effect in inhibiting cortical excitability in general, its inhibition effect has been commonly found when stimulating motor areas [38]. Moreover, it appeared to effectively downregulate the excitability of the cerebellum [8,35]. Interestingly, previous studies have not produced a consistent cathodal ctDCS effect in sensorimotor adaptation [12,13,15,16,39,40]. Thus, we included cathodal ctDCS in Experiment 2, in addition to the anodal and sham ctDCS used in Experiment 1. The same implicit adaptation enabled by the error clamp learning was used, but this time we used a smaller clamp angle of 4° as opposed to 30°. Clamp angles smaller than 6° would lead to smaller implicit adaptation [41], thus leaving some room for potential improvement by ctDCS stimulation. Different from Experiment 1, Experiment 2 involved three phases of trials completed on a single day (Figure 1C). Only four target locations were used (45°, 135°, 225°, and 315°); thus, a trial cycle only had 4 trials. The experiment consisted of 200 cycles distributed in three phases, i.e., baseline (40 cycles), learning (80 cycles), and decay (80 cycles). Compared to Experiment 1, the learning phases increased from 40 cycles to 80 cycles since the implicit adaptation to 4° rotation was considerably slower. However, as each cycle reduced its trial number down to four, the duration of the adaptation phase remained approximately 20 min, as in Experiment 1. In the first 30 cycles of the baseline phase, the cursor was visible and veridical. In keeping with Experiment 1, in the next 10 baseline cycles, the cursor feedback was withdrawn and the ctDCS stimulation was started. In the learning phase, the cursor was clamped at 4° CCW towards the target, and the ctDCS was on all the time during the adaptation phase. In the decay phase, there was no feedback. The three groups of participants received anodal, cathodal, or sham ctDCS, respectively. Experiment 2 lasted approximately 1.5h, including the experiment instructions and preparation of tDCS.

#### 2.4.3. Experiment 3: Implicit Adaptation to Gradual Perturbations

Instead of eliciting implicit adaptation to the task-irrelevant clamped error, Experiment 3 employed another technique to elicit implicit adaptation to the task-relevant error. We used a 30° visuomotor rotation relative to the actual hand movement, but the rotation was not introduced abruptly but gradually from 0. This gradually increasing perturbation prevents the participant from noticing the perturbation and thus makes implicit adaptation prominent [42]. Similar to Experiment 1, Experiment 3 consisted of data collection on two consecutive days, each of which had three phases of trials (Figure 1C). On day 1, the baseline phase included 20 cycles of trials with veridical feedback. In the adaptation phase, the CCW rotation was gradually imposed from 0° to 30° in 40 cycles (320 trials) with a rate of 0.0938° per trial. In keeping with Experiments 1 and 2, ctDCS was applied only on day 1, turned on starting from the mid of the baseline phase (cycle 11), and lasted 20 min, which approximately covered the adaptation phase. The decay phase consisted of 40 cycles of no-feedback trials. On day 2, the procedure was identical to that of Experiment 1: After the baseline phase, the cursor was abruptly rotated 30° CCW and maintained for 40 cycles. Thus, we could examine whether cerebellar ctDCS could exert its influence via another form of implicit adaptation, i.e., gradual adaptation. Participants were randomly assigned to an anodal or sham group and performed a total of 110 cycles (880 trials). The duration of Experiment 3 was a similar 1.5 h as in Experiment 1.

### 2.5. Data Analysis

The position of the digital pen, representing the hand position, was registered at a frequency of 125 Hz for offline analysis. The positional data were low-pass filtered at 10 Hz with a 4-th order Butterworth filter and then numerically differentiated to calculate the velocity. Since the adaptation was to counter a visuomotor rotation, the changes in the direction of the shooting movement were indicative of learning. The angular deviation (AD), the directional deviation of the actual movement from the desired direction towards the target, was thus taken as the performance variable. The movement direction was defined as the direction of the vector spanning from the starting position to the position of the hand at the peak outward velocity. A positive angular deviation indicated a clockwise error, whereas a negative one indicated a counterclockwise error. We averaged angular deviation over 8 (Experiments 1 and 3) or 4 (Experiment 2) trials in a trial cycle to reflect the learning changes in a cycle. The onset of each movement was determined as the time point at which the radial velocity exceeded 5% of its peak velocity. The reaction time (RT) was thus operationally defined as the time difference between the target appearance and movement onset, and movement time (MT) was defined as the time difference between the movement onset and when the radial distance of movement exceeded 5 cm.

To characterize the adaptation performance, we computed the adaptation rate, extent, and retention effect, including aftereffect, cross-day retention, and re-adaptation rate. The AD of the last five cycles of the baseline phase was averaged to account for the directional bias for each individual participant; this bias was deducted from the AD from the subsequent trials. To compute the adaptation rate, we first determined the window of trial cycles covering the initial adaptation by examining the adaptation to visuomotor rotation in Experiment 1, which had the largest sample size. Paired t-tests were used to locate the cycle in which AD stopped to further increase from the immediately preceding cycle [43]. The anodal group stopped the rapid adaptation at cycle 7, while the sham group was at cycle 11. We thus took a window size of 9 cycles and computed the average AD over these 9 cycles as an indicator of the adaptation rate. This window size was similar to what had been reported in previous studies on visuomotor rotation [9,42]. The same window size was used to measure the initial adaptation rate of Experiments 1 & 2 and the re-adaptation rate of Experiments 1 & 3. In Experiment 3, the initial adaptation was to correct a gradually imposed rotation, and the AD would increase monotonically; therefore, using a fixed window was no longer appropriate. We thus calculated the averaged trial-by-trial learning rate by dividing the AD change from one trial to the next by the performance error experienced in that trial. We averaged this single-trial learning rate over all the trials as the adaptation rate for gradual perturbation in Experiment 3 [44,45,46,47]. For all the experiments, adaptation extent was defined as the averaged AD of the last 10 cycles of the adaptation or the re-adaptation phase. In the decay phase, the adaptation passively decayed without feedback. The first cycle was the aftereffect of the adaptation. The decay rate was estimated by fitting the data in the decay phase with an exponential function x(m) = a^ (m − n) × (n) where x(m) and x(n) were the ADs for the m th and n th trial, and the parameter a denoted the decay rate. Because the decay data from each individual participant was noisy for fitting the exponential, we bootstrapped each group of participants 1000 times and computed the decay rate each time. Thus, the group difference can be analyzed by comparing the 95% confidence interval of the decay rate. The last 10 cycles of the decay phase were averaged to quantify the decay residue. In Experiments 1 and 3, the participants were also tested for re-adaptation to visuomotor rotation with a task-relevant cursor on day 2. The AD in the first cycle on day 2 was used as an indicator of cross-day retention. Note that here we used the baseline from day 1 since the cross-day retention was evaluated. Re-adaptation rate, re-adaptation extent, decay rate, and decay residue were calculated using the same methods as day 1.

## 3. Results

### Experiment 1: No Evidence of Anodal ctDCS Effect on Error Clamp Adaptation

On day 1, we found clear implicit adaptation to the error clamp in both the anodal and sham groups. Both groups exhibited a gradual build-up of hand deviations despite them being instructed to aim for the target (Figure 2A). When the error clamp perturbation was removed, the AD gradually decreased towards 0°, indicating a slow decay of implicit adaptation. On day 2, the participants exhibited cross-day retention as both groups had significant AD (larger than 0°) in the first cycle of the baseline phase (anodal: 3.98 ± 3.56°, sham: 4.22 ± 3.77°, both *p*s < 0.001). This first-cycle retention was quantitatively similar to what had been left off in the previous day, as it did not differ statistically from the last cycle of the decay phase in day 1 (*t* (81) = 1.49, *p* = 0.14). After the baseline phase, an abrupt 30° visuomotor rotation was applied to the cursor, and the participants were required to hit the target with the task-relevant cursor. Like previous studies, the participants quickly re-adapted to the visuomotor rotation, presumably aided by strategic learning. The involvement of explicit learning was evidenced by an immediate increase in reaction time upon receiving the rotation perturbation [25,26] and its subsequent gradual decrease over the period of adaptation (Figure 2B).

Though both groups showed typical implicit adaptation, they did not differ in any metrics of the initial adaptation and its subsequent retention. In the order of the adaptation process, we compared the measures that characterized initial adaptation performance including initial adaptation rate and extent on day 1, as well as the measures that characterized adaptation retention including decay rate and residue on day 1, and cross-day retention, re-adaptation rate and extent, decay rate and residue on day 2 (summarized in Table 1). We also checked reaction time, whose abrupt increase indicated the usage of explicit strategy. The change of reaction time in the initial learning window, relative to the 5 cycles of the baseline phase, was compared between groups. But we did not observe significant group differences either (Table 2), indicating that the anodal ctDCS applied during implicit adaptation did not facilitate explicit learning. None of these measures differed between the two groups. In fact, all the Bayesian factors were smaller than 1/3, suggesting moderate support for a null effect.

In summary, we were able to elicit typical implicit adaptation to the error clamp and re-adaptation of visuomotor rotation but did not find evidence for the effect of anodal ctDCS during adaptation and retention.

The implicit adaptation to the error clamp saturates with increasing rotation size beyond 6°, below which the adaptation rate increases with rotation size [41]. Thus, one possible confound that caused the ineffectiveness of ctDCS in Experiment 1 was that the implicit learning induced by the error clamp with the large 30° rotation saturated, and this might have left no room for ctDCS-related improvement. To rule out this possible ceiling effect, Experiment 2 used a small rotation angle of 4°, and also added another stimulation group which received cathodal ctDCS. Again, all groups exhibited a slow, gradual adaptation to the error clamp and slowly returned to the baseline during the decay phase (Figure 3). We also confirmed that the adaptation rate was significantly slower with the 4° rotation by comparing the average adaptation rate across all groups from Experiment 2 to that from Experiment 1 (*t* (118) = 7.08, *p* < 0.001). Thus, with a smaller error clamp, the implicit adaptation was slower than the maximum implicit adaptation rate that was allowed. Importantly, we did not find any group difference in all the adaptation performance measures (Table 2). Note, we did not collect the re-adaptation data to assess cross-day adaptation, but within-day retention, as revealed by the data in the decay phase, failed to show any group difference. All the Bayesian factors were less than 1/3, suggesting moderate support for the null effect.

Implicit adaptation induced by error clamp involves no control of the cursor, but the traditional sensorimotor adaptation demands an active recalibration of the sensorimotor system to accommodate perturbations. To examine whether implicit adaptation based on active recalibration is affected by ctDCS, we employed a gradually imposed visuomotor rotation, which has been shown to have a minimum contribution from explicit learning [42,48,49]. Experiment 3 had identical procedures as Experiment 1 except that the initial adaptation on day 1 was for gradual perturbations instead of for error clamp. We found that the participants actively compensated for the gradual perturbation by rotating their movement direction to follow the linearly increasing perturbations (Figure 4A). Only in the later phase of the adaptation phase, the hand “lagged” behind the increasing perturbations. The fact that they did not nullify the error in this late learning phase indicated that they did not form an explicit strategy to counter the perturbations [50]. In the decay phase when the rotation perturbation was removed, the hand angle slowly decayed towards 0°. On day 2, the participants adapted to the 30° visuomotor rotation. We found their re-adaptation rate was similar to that of the participants in Experiment 1, who also experienced implicit adaptation based on the error clamp. The fact that there was no difference between these two experiments (comparing the re-adaptation rate from all the participants in these two experiments, *t* (134) = 0.58, *p* = 0.56) supported that the adaptation to gradual perturbations was largely implicit; otherwise, any strategic learning in day 1 would lead to faster re-adaptation in day 2, i.e., exhibiting a saving effect [42].

Importantly, we again did not find evidence to support a ctDCS effect on implicit adaptation. Similar to Experiment 1, we compared all the adaptation measures between the anodal and sham groups (Table 3). The initial adaptation rate and extent were defined differently here: an average learning amount in a cycle was computed to track the “linear” adaptation to gradually imposed visuomotor rotation (Figure 4A). The initial adaptation extent was defined as the average AD at the end of the adaptation phase. No significant difference was found between the anodal and sham groups in adaptation rate or extent. The only group difference is that on day 1 the anodal group had a faster decay rate (95% CI, anodal: 16.10 ± 3.41°, sham: 17.00 ± 3.67°) and less decay residue (anodal: 3.90 ± 2.92°, sham: 5.99 ± 3.44°, *t* (52) = 2.41, *p* = 0.02). However, this weak ctDCS effect on adaptation retention on day 1 was not carried over to day 2. Their cross-day retention, estimated by the first cycle on day 2, did not differ between groups. Their re-adaptation to visuomotor rotation and subsequent decay also did not differ between groups (Table 3). We examined the change of reaction time when the perturbations were abruptly applied as an indicator of explicit strategy. Though both anodal and sham groups exhibited a longer reaction time, their RT change relative to the baseline did not differ. In summary, we did not find adequate evidence that anodal ctDCS affected implicit learning enabled by task-relevant feedback and active error correction, despite a weak effect on the adaptation decay in day 1.

## 4. Discussion

The effect of cerebellar tDCS on sensorimotor adaptation has been inconsistently reported, and the current debate is whether the relative contribution of the explicit and implicit components during adaptation causes the inconsistency. The present study directly tested sensorimotor adaptation tasks that are widely accepted as implicit and free of explicit learning. Experiment 1 used error-clamp feedback to induce implicit adaptation and found that anodal ctDCS failed to improve the adaptation rate or adaptation retention within a day or across days. As a follow-up, Experiment 2 examined a smaller error clamp adaptation to rule out a possible ceiling effect in Experiment 1 and found that both anodal and cathodal ctDCS failed to produce changes in implicit adaptation. Experiment 3 examined another form of implicit adaptation with gradually induced visuomotor rotation but again did not find evidence for the effect of anodal ctDCS. Our findings highlight that sensorimotor adaptation is not affected by ctDCS applied during the initial adaptation even when implicit learning dominates adaptation, and the reported inconsistency in the literature should be re-examined by taking explicit learning processes into consideration.

Our experiments replicated behavioral patterns of implicit adaptation [32,41,42,48,49,51], including a gradual but saturated adaptation and an ensuing slow decay after error clamp adaptation (Experiments 1 and 2), cross-day retention if the adaptation on day 1 was not completely washed away (Experiment 1), a faster adaptation with the 30° than the 4° error clamp (Experiment 1 vs. 2), a fast re-adaptation to an abrupt visuomotor rotation on day 2 (Experiments 1 and 3), a monotonically increasing adaptation to gradual perturbations (Experiment 3). All the participants were not aware of their adaptation since they intended to move straight towards the target but unknowingly rotated their movement direction (Experiments 1 and 2) or failed to move the cursor towards the target despite a large performance error at the late phase of gradual adaptation (Experiment 3). Importantly, we did not observe any effect of ctDCS on implicit adaptation after thoroughly examining the common performance measures during and after the adaptation. The two recent studies on visuomotor rotation implied that the conditions with more implicit learning involved tended to be positively affected by anodal ctDCS [29,30]. Our findings do not agree with the hypothesis that the relative contribution of implicit learning is the determinant for the ctDCS effect on sensorimotor adaptation. With a combined sample size of 174 participants, our results provide convincing evidence that ctDCS applied during implicit adaptation cannot modulate apparent adaptive behaviors.

Though sensorimotor adaptation was traditionally viewed as a testbed for examining procedural learning, recent studies have shown that various adaptation paradigms involve both explicit and implicit learning. Besides the visuomotor rotation investigated here, reaching adaptation with force field [52] or prism goggles [53], walking adaptation with a split-belt [54], and saccadic adaptation with a target jump [55] all involve cognitive components and strategic corrections to the perturbation. Taking force field adaptation as an example, people rely on proprioceptive feedback to adapt to a novel mechanical environment and have difficulty verbalizing their adaptation solution. Thus, early theorization of its neural basis focused on the cerebellum [56]. However, recent studies revealed that people also developed an explicit strategy to counter the force field perturbation [52]. Thus, since the experimental paradigms invoked explicit learning [28], all the previous ctDCS studies actually investigated the combined effect of explicit and implicit learning, as opposed to studying procedural or implicit learning alone as originally intended. Using the error clamp and gradual perturbation paradigms, our study shows that ctDCS has no effect when implicit adaptation is the sole driver for sensorimotor adaptation.

What do our findings contribute to the inconsistency in the effect of ctDCS on sensorimotor adaptation if the relative contribution of implicit learning is not the underlying cause? We postulate that large inter-individual differences and diversity in task specifics contribute to the inconsistency among previous studies. We noticed that nearly all studies reporting a positive effect showed an increased initial adaptation rate with anodal ctDCS [9,10,11,13,30,57]. Take the mostly used paradigm visuomotor rotation as an example. The initial adaptation rate is typically fast owing to the large contribution of explicit learning that is formed during early adaptation, which is also shown in our data on day 2 [58]. The formation and utilization of explicit strategy can be achieved in a few trials [59], but its timing of formation and size have large inter-individual differences [22,58]. It is thus expected that a biased sample of “faster” explicit learners would show accelerated initial adaptation, a problem that could be exacerbated by the publication bias for positive results in neural stimulation studies [60]. Furthermore, many task specifics, including reaction time [25,26], participants’ working memory capacity [24] and age [27], prior instruction about the perturbation [23], and the orientation of the display [29] can all affect the formation of explicit strategy, thus contributing to the large variance between studies. Although limited testing has been done, large variability in explicit learning is expected in adaptation paradigms other than visuomotor rotation, contributing to the inconsistency of the ctDCS effect. Our experiments showed that after removing the variable contribution of explicit learning, the ctDCS has no effect on sensorimotor adaptation, especially for the initial adaptation rate.

Whether cerebellar tDCS can affect explicit learning should be investigated further, given the advance in neurophysiology and functionality of the cerebellum. There is accumulating evidence that the cerebellum is involved in various high-level cognitive functions, including decision-making [61], reasoning [62], linguistic function [63], and social cognition [64]. Cerebellar lesions can impair execution functions, including planning, set-shifting, abstraction, working memory, and verbal fluency [65]. There is evidence that distinct cerebellum areas support explicit and implicit learning in sensorimotor adaptation. Patients with posterior cerebellar lesions show impaired initial adaptation, which is believed to be explicit learning-dominated, but intact aftereffects, which is implicit learning-dominated [66]. In contrast, patients with superior cerebellar lesions show deficits in aftereffects, suggesting the role of the superior cerebellum in implicit processes in sensorimotor adaptation [67]. Neural imaging studies on visuomotor rotation and prism adaptation also found that distinct areas of the cerebellum were associated with explicit and implicit learning [68,69,70]. Relevant to the visuomotor rotation adaptation here, recent behavioral and modeling work has proposed that explicit learning is driven by performance error while implicit learning is driven by sensory prediction error [71]. Performance error is the difference between movement outcome and goal, and the sensory prediction error is the difference between the predicted outcome of one’s movement and real feedback. Interestingly, both types of errors are represented in the cerebellum [72,73]. Hence, the cerebellum is relevant for both explicit and implicit learning. However, whether ctDCS can directly affect explicit learning has not been studied except for the null result reported by Liew et al. [29]. They asked participants to verbally report their aiming direction before each movement as a measure of explicit learning. However, the aiming report not only shows large variance within and across participants [74] but also promotes strategy use, i.e., increases explicit learning itself [75]. Thus, their null effect might result from or at least be affected by a possible ceiling effect induced by the aiming report. In future studies, other measures of explicit learning should be used in order to get a more unbiased estimate, including so-called exclusion trials [i.e., requiring the participants to directly move to the target without an aiming strategy and without feedback; 75]) or eye fixations before the movement [76]. Both methods are able to quantify the amount of explicit learning [75,76] but have not been used in combination with ctDCS paradigms. Our study isolated implicit learning and examined the putative ctDCS effect; future investigations should seek novel experimental manipulations to isolate explicit learning in sensorimotor adaptation and examine how the cerebellum contributes to this cognitive component of procedural learning.

Besides the possibility that ctDCS affects explicit learning alone, the possibility that ctDCS affects both explicit and implicit learning simultaneously or even their interaction cannot be excluded. Since the cerebellum has dual representations of performance error [72] and sensory prediction error [73], the two teaching signals for explicit and implicit learning, respectively [71], ctDCS might affect both types of learning simultaneously. As the excitability modulation by tDCS lacks spatial precision [77], it is hard to predict how the neural substrate for two types of learning is affected by its diffusive stimulation. To make the matter more complicated, the interaction between explicit and implicit learning is currently under debate: while early models view them as independent processes with their summative effect driving adaptation [21,22], recent work has shown that they can also compete for the same performance error [78], or work in tandem so that implicit learning compensates for the errors that explicit learning cannot correct [79]. Without a mechanistic view of the interaction between the two learning components and precisely localizing the stimulation on the cerebellum where both error signals reside, it is still premature to predict the effect of ctDCS on sensorimotor adaptation.

Our findings also have implications for the theorization of implicit adaptation. The predominant view is that error clamp adaptation is driven by a single process based on sensory prediction error [32,41]. Here, the sensory prediction error is defined as the angular difference between the predicted or intended straight movement towards the target and the to-be-ignored clamped cursor movement (Figure 1B). However, a recent study suggested that this error can also be viewed as a performance error, i.e., the motor system mistakes the clamped cursor as self-controlled, and thus, the deviation of the cursor from the aiming target can be viewed as a performance error [78]. While this ongoing debate is about the semantic meaning of an error, it is noteworthy that both accounts view the error clamp adaptation as being driven by a single error, either being a sensory prediction error or performance error. Our null results appear to suggest that ctDCS did not affect the error processing during implicit adaptation. An intriguing possibility is that both types of errors may drive the implicit adaptation induced by the error clamp; thus, the aforementioned unresolved interaction between them might cause the null effect of ctDCS, similar to other inconsistent reports.

## 5. Conclusions

In conclusion, our study employed novel experimental manipulations to isolate implicit adaptation and revealed that cerebellar tDCS failed to modulate adaptation and its retention within and across days. With a large sample size and extensive examination of performance measures, we provided solid evidence that diffusive stimulation of the cerebellum cannot lead to observable changes in implicit sensorimotor adaptation. The proposition that the more implicit learning is involved in sensorimotor adaptation, the more likely the effect of cerebellar tDCS is not supported. Instead, our findings suggest that the complex interplay between cognitive/explicit learning and procedural/implicit learning and the large individual differences associated with this interplay might contribute to the inconsistent findings from previous studies on cerebellar tDCS and sensorimotor adaptation. Given the dual representations of error signals supporting the two types of learning in the cerebellum [72,73], future investigations should utilize more precise noninvasive brain stimulation tools such as navigated transcranial magnetic stimulation [80] to pinpoint the sensorimotor or cognitive regions in the cerebellum [65], in addition to novel behavioral experiment designs that are able to isolate different learning components. 

## Figures and Tables

**Figure 1 brainsci-12-01325-f001:**
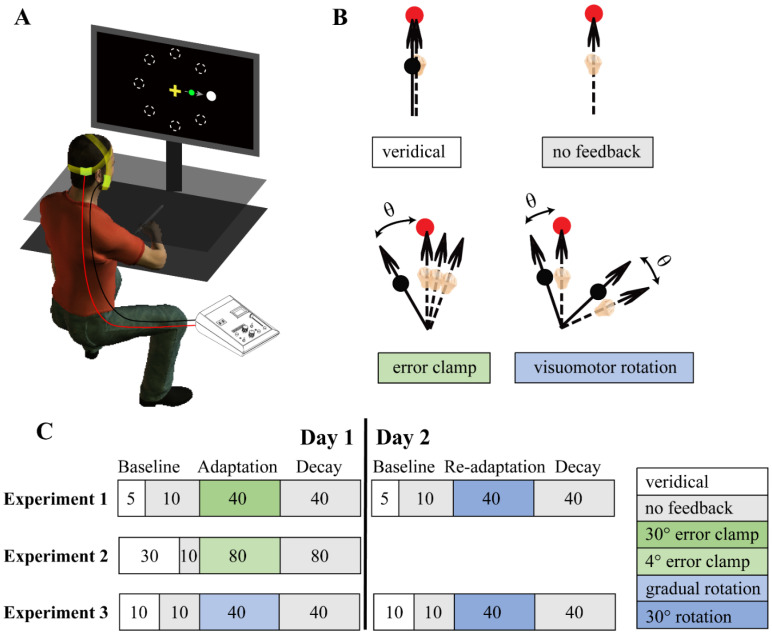
Schematic illustrations of the experimental setup (**A**), movement tasks (**B**), and procedures (**C**). (**A**) The seated participant, wearing tDCS electrodes, moves the hand from a center position to an outward target with the guidance of a vertical display. The hand motion is concealed by a blinder placed at the shoulder level and tracked by the hand-held digital pen sliding on a tablet. (**B**) Illustrations of the four trial types used in the experiments. Veridical trials show a cursor whose position is overlaid with the tip position of the hand-held pen. Trials with no feedback do not show the hand cursor during movement. Trials with the error clamp show a cursor moving concurrently with the hand but with an angular deviation. The angular deviation is a fixed counterclockwise rotation angle θ of 30 or 4° related to the target, depending on the experiment. The unseen hand slowly biases in the clockwise direction as a result of implicit adaptation to the task-irrelevant cursor. The trials with visuomotor rotation show a concurrent cursor whose movement direction is controlled by the actual hand direction but with a counterclockwise rotation angle θ. Differently from the error clamp, this task-relevant cursor’s direction depends on the hand direction (θ unchanged related to the hand, as shown), and it leads to adaptive changes as the hand moves in the clockwise direction to offset the rotation. (**C**) The illustration of procedures for the three experiments. Each session, either collected in one day or in two consecutive days, has three experimental phases, i.e., the baseline, adaptation (or re-adaptation), and decay phases. The number of trials in each phase, coded by color shades, is also shown for each experiment.

**Figure 2 brainsci-12-01325-f002:**
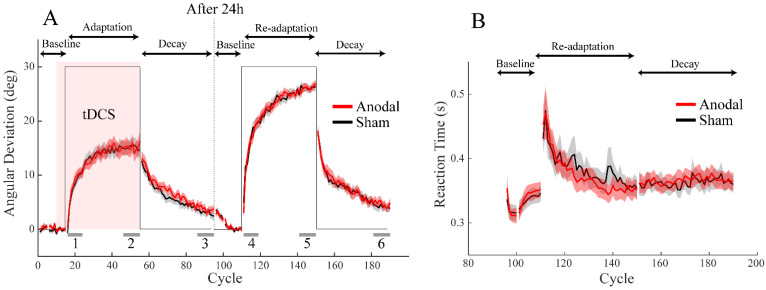
Adaptation performance in Experiment 1. (**A**) The angular deviation as a function of trial cycles over two days. The group average (line) and standard error across participants (shade) are shown. The anodal and sham groups are plotted in red and grey, respectively. The experimental phases are marked by arrows on top of the figure. The red rectangular area denotes the duration of the ctDCS application. The solid black line denotes the error clamp/rotation perturbation. The grey segments on the bottom of the figure show the windows for calculating performance measures: 1 and 4 are for the adaptation and re-adaptation rate, respectively; 2 and 5 are for the adaptation and re-adaptation extent, respectively; 3 and 6 are for the decay residues in the two days. (**B**) Reaction time as a function of trial cycles for the baseline and the re-adaptation phases on day 2. There was a marked increase in RT when the visuomotor rotation was unexpectedly applied.

**Figure 3 brainsci-12-01325-f003:**
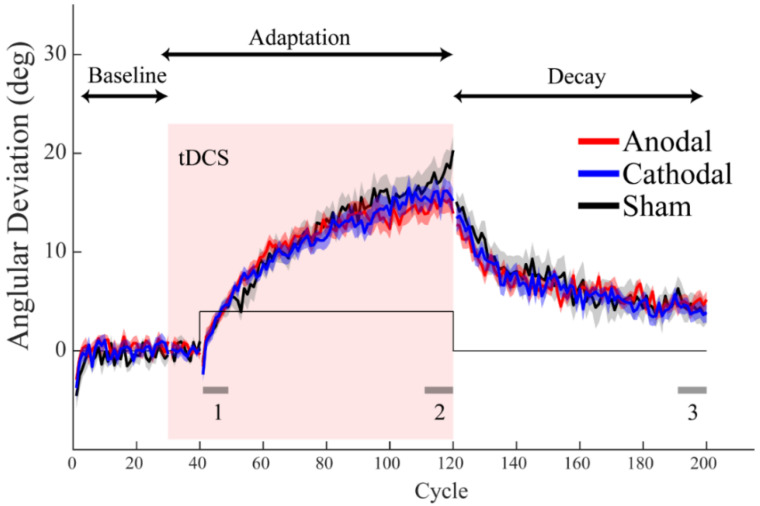
Adaptation performance in Experiment 2. The angular deviation as a function of trial cycles. The group average (line) and standard error (shade) across participants are shown. The anodal, cathodal, and sham groups are plotted in red, blue, and grey, respectively. Only the baseline, adaptation, and decay phases within a day were collected and marked by arrows on top of the figure. The red rectangular area denotes the duration of ctDCS application. The solid black line denotes the rotation perturbation. The grey segments on the bottom of the figure show the windows for calculating performance measures: 1 to 3 are for adaptation rate, adaptation extent, and decay residue, respectively.

**Figure 4 brainsci-12-01325-f004:**
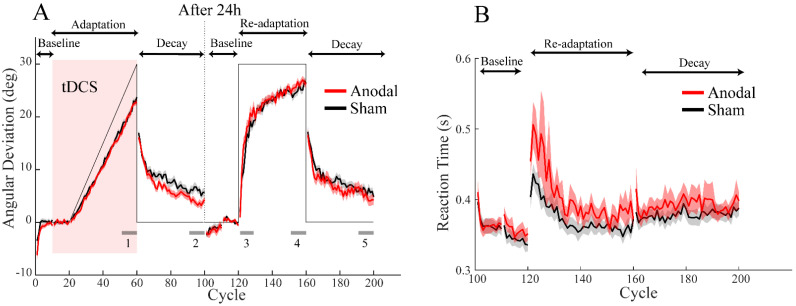
Adaptation performance in Experiment 3. (**A**) The hand deviation angle as a function of trial cycles over two days. The group average (line) and standard error (shade) across participants are shown. The anodal and sham groups are plotted in red and grey, respectively. The experimental phases are marked by the arrows on top of the figure. The red rectangular area denotes the duration of the ctDCS application. The solid black line denotes the visuomotor rotation perturbation. The grey segments on the bottom of the figure show the window for calculating performance measures: 1 and 4 are for the adaptation and re-adaptation extent, respectively; 2 and 5 are for decay residues in the two days; 3 is for the re-adaptation rate on day 2. (**B**) Reaction time as a function of trial cycles for the baseline and the re-adaptation phases on day 2. There is a marked increase in RT when the visuomotor rotation is unexpectedly applied.

**Table 1 brainsci-12-01325-t001:** Group comparisons of performance measures in Experiment 1.

		Anodal	Sham	*t* (80)	*p*-Value	Bayesian Factor
Day 1	Initial adaptation rate (mean ± SD)	7.83 ± 4.28°	7.31 ± 4.10°	0.56	0.58	0.26
	Initial adaptation extent (mean ± SD)	15.14 ± 9.25°	15.09 ± 8.23°	0.028	0.98	0.23
	Decay rate (95% CI)	(0.958, 0.971)	(0.950, 0,970)			
	Decay residue (mean ± SD)	3.84 ± 4.02°	3.09 ± 3.82°	0.88	0.38	0.32
Day 2	Cross-day retention (mean ± SD)	3.98 ± 3.56°	4.22 ± 3.77°	0.31	0.75	0.24
	Re-adaptation rate (mean ± SD)	14.91 ± 5.77°	15.27 ± 5.57°	0.16	0.77	0.23
	Re-adaptation extent (mean ± SD)	26.09 ± 4.11	26.05 ± 4.45	0.66	0.96	0.28
	Decay rate (95% CI)	(0.964, 0.975)	(0.956, 0.975)			
	Decay residue (mean ± SD)	4.72 ± 4.90°	4.22 ± 4.75°	0.91	0.64	0.33
	Reaction time (mean ± SD)	365 ± 87 ms	366 ± 88 ms	0.36	0.72	0.24
	ΔRT (mean ± SD)	73 ± 92 ms	76 ± 114 ms	0.030	0.98	0.23

Experiment 2: No evidence of anodal or cathodal ctDCS effect on error clamp adaptation with a small clamp size.

**Table 2 brainsci-12-01325-t002:** Group comparisons of performance measures in Experiment 2.

	Anodal	Cathodal	Sham	*F* (2,32)	*p*-Value	Bayesian Factor
Initial adaptation rate (mean ± SD)	2.71 ± 1.80°	2.49 ± 1.81°	2.57 ± 1.63°	0.055	0.95	0.13
Initial adaptation extent (mean ± SD)	14.75 ± 4.30°	15.62 ± 3.91°	17.37 ± 5.40°	1.00	0.24	0.22
Decay rate (95% CI)	(0.985, 0.991)	(0.979, 0.988)	(0.966, 0.989)			
Decay residue (mean ± SD)	4.81 ± 2.52°	4.44 ± 2.55°	4.75 ± 4.22°	0.0579	0.85	0.14

Experiment 3: No evidence of anodal ctDCS effect on implicit adaptation induced by gradual perturbations.

**Table 3 brainsci-12-01325-t003:** Group comparisons of metrics for implicit adaptation in Experiment 3.

		Anodal	Sham	*t* (52)	*p*-Value	Bayesian Factor
Day 1	Initial adaptation rate (mean ± SD)	0.16 ± 0.067	0.17 ± 0.080	0.60	0.55	0.32
	Initial adaptation extent (mean ± SD)	20.31 ± 1.46°	20.67 ± 2.54°	0.64	0.53	0.32
	First cycle of decay (mean ± SD)	16.10 ± 3.41°	17.00 ± 3.67°	0.93	0.36	0.39
	Decay rate (95% CI)	(0.952, 0.968)	(0.969, 0.980)			
	Decay residue (mean ± SD)	3.90 ± 2.92°	5.99 ± 3.44°	2.41	0.020	2.82
Day 2	Cross-day retention (mean ± SD)	−0.26 ± 2.77°	−0.29 ± 3.27°	0.04	0.47	0.27
	Re-adaptation rate (mean ± SD)	15.51 ± 4.73°	13.59 ± 4.35°	1.55	0.13	1.15
	Re-adaptation extent (mean ± SD)	25.96 ± 3.21°	25.16 ± 3.26°	0.91	0.37	0.56
	Decay rate (95% CI)	(0.971, 0.983)	(0.970, 0.983)			
	Decay residue (mean ± SD)	5.06 ± 4.78°	5.87 ± 3.78°	0.69	0.49	0.29
	Reaction time (mean ± SD)	389 ± 65 ms	371 ± 47 ms	1.24	0.22	0.51
	ΔRT (mean ± SD)	123 ± 201 ms	68 ± 39 ms	1.38	0.17	0.60

## Data Availability

The data can be found at https://github.com/HJWangOrder66/Motor-learning-with-tDCS.

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
