# Peer review of "Transcranial Direct-Current Stimulation Does Not Affect Implicit Sensorimotor Adaptation: A Randomized Sham-Controlled Trial"

_brainsci, 2022, doi:10.3390/brainsci12101325_

Round 1

Reviewer 1 Report

Thank you for giving me the opportunity to review this manuscript.

This study demonstrated that both anodal and cathodal tDCS over cerebellum did not affect implicit sensorimotor adaptation.

I think that it is important to publish results of randomized controlled trials to avoid publication bias. Furthermore, the discussion section was well written. However, it is necessary to revise this manuscript.

-major

1)     Please describe the study design more clearly. I think Experiment 1, 2, and 3 were all randomized controlled trials. Furthermore, please describe how random sequence was generated, allocation was concealed, whether tDCS administrators were blinded, and whether this study protocol was pre-registered in Experiment 1, 2, and 3, respectively.

2)     Please describe baseline characteristics of anodal tDCS, cathodal tDCS and sham tDCS arms, respectively.

3)     I think that tDCS protocol in this study was not perfect. How tDCS stimulates cerebellum was unclear. First, please explain how tDCS was placed over cerebellum, and how you find any places to target cerebellum (10-20 EEG methods?). Second, one session tDCS was usually too short-lasting week effects. How long does it take to stimulate cerebellum by tDCS? Third, 2mA anodal tDCS over cerebellum seemed to be weak. That is because tDCS effects were usually too diffused and affected by the impedance of skull. How was it possible to enhance cortical excitability of cerebellum by anodal tDCS over cerebellum? Fourth, I think the effects of cathodal tDCS on depolarization were still controversial. Therefore, it is difficult to down-regulate cerebellar excitability by cathodal cerebellar tDCS in the second experiment. Why did you choose tDCS rather than rTMS in this study? Please explain all of them in the manuscript.

-minor

1)     Please change the title of “Modulating cerebellar excitability by transcranial direct-current stimulation does not affect implicit sensorimotor adaptation” as “transcranial direct-current stimulation does not affect implicit sensorimotor adaptation; A randomized sham-controlled trial” That is because whether tDCS modulated cerebellar excitability was unclear.

2)     In line 256, please delete the sentence of “In Experiment 2, we upregulated and downregulated the excitability of the cerebellum by using anodal and cathodal ctDCS, respectively.”, because it is not always possible. In line 534, please delete the sentences of “Whether cerebellar tDCS can affect explicit learning should be investigated further, given the advance in neurophysiology and functionality of the cerebellum.” That is because this study showed no effects of tDCS on those learning in this population. Please change the sentences as, for example, “Whether cerebellar non-invasive brain stimulation (other than tDCS) can affect explicit learning should be investigated further, given the advance in neurophysiology and functionality of the cerebellum.”

3)     Please delete the sentences of “teaching signals for explicit and implicit learning, respectively, ctDCS might affect both types of learning simultaneously.” That is because this study showed ctDCS did not show any types of learning.

4)     In line 291, should “cerebellar ctDCS” be “ctDCS”?

5)     Please delete the sentences of “Our findings do not support either account: given that both types of errors are encoded by the cerebellum[69, 70], it is expected that ctDCS could modulate this implicit adaptation if it is driven by one of them.”(Line 592-594).  I agree with your point in the conclusion that “future investigations should utilize more precise non-invasive brain stimulation tools”(other than tDCS). I think that HD-tDCS is still not precise. That is because the effects of tDCS is not only diffuse but also too superficial. Furthermore, another study showed that “tDCS did not explicit and implicit learning during visuomotor adaptation.” (Liew, S.-L., et al. Frontiers in Neuroscience, 2018. 12: p. 610.). What kind of non-invasive brain stimulation tools should be utilized?

6)     Although previous study showed that “Anodal tDCS elicited an increase in the magnitude and duration of motor memories in a polarity-specific manner, as reflected by changes in the kinematic characteristics of TMS-evoked movements after anodal, but not cathodal or sham stimulation.”, it is impossible to make a conclusion that brain polarization enhances the formulation and retention of motor memories by anodal tDCS (Galea, J.M. and P. Celnik, Journal of Neurophysiology, 645 2009. 102(1): p. 294-301.). First, the sample size of this study was too small. Second, the effects of anodal tDCS on motor memories were assessed by TMS-evoked movements. How should future studies assess the effects of motor memories after non-invasive brain stimulation?

7)     I agree with your point that “In future studies, other measures of explicit learning should be used in order toget a more unbiased estimate, including so-called exclusion trials”. What kind of outcomes measures should be utilized to detect differences?

I think it is necessary to revise the manuscript.

Author Response

Dear reviewers,

Thank you very much for your positive and constructive comments and suggestions, which have helped us dramatically improve the manuscript. In the revisions, we have given more explanations about the rationale of our experimental methodology, toned down some conclusions, and discussed relevant issues extensively. We also went through the writing to correct some grammar mistakes. We believe the current manuscript has adequately addressed all your previous concerns; please see our point-to-point responses in the attachment.

Reviewer 2 Report

Reading the manuscript written by Wang and Wei, was really interesting. The study aimed to evaluate the ctDCS effect on sensorimotor adaptation using error clamp perturbation and gradual perturbation, two effective techniques to elicit implicit learning only. Their results suggest that the complex interplay between the two learning processes and the large individual differences associated with this interplay might contribute to the inconsistent findings from previous studies on cerebellar transcranial direct current stimulation (ctDCS) and sensorimotor adaptation.

This article is written in a concise and orderly manner. The methodologies are appropriate and aligned with the proposed objectives. The images and figures used are very suggestive and of a superior quality. Very well-chosen statistical analysis methods. The massage from this manuscript is quite meaningful. The article is easy to read, and can be of interest to readers and researchers. The English is good, but there are spelling, punctuation and some grammar issues (sometimes the spaces between the words are missing, other times there are too many). This will apply to the whole manuscript.

Please specify the Author Contributions.

Author Response

(The authors gave the same response as above.)

Round 2

Reviewer 1 Report

Thank you for revising the manuscript.

I think this manuscript would be suitable for publication in this journal.